# DISCRETE STRUCTURAL PLANNING FOR GENERATING DIVERSE TRANSLATIONS

## ABSTRACT

Planning is important for humans when producing complex languages, which is a missing part in current language generation models. In this work, we add a planning phase in neural machine translation to control the global sentence structure ahead of translation. Our approach learns discrete structural representations to encode syntactic information of target sentences. During translation, we can either let beam search to choose the structural codes automatically or specify the codes manually. The word generation is then conditioned on the selected discrete codes. Experiments show that the translation performance remains intact by learning the codes to capture pure structural variations. Through structural planning, we are able to control the global sentence structure by manipulating the codes. By evaluating with a proposed structural diversity metric, we found that the sentences sampled using different codes have much higher diversity scores. In qualitative analysis, we demonstrate that the sampled paraphrase translations have drastically different structures.

## 1 INTRODUCTION

When humans speak, a planning phase exists when we want to say in a particular structure or style. Linguists have found particular speech errors or behaviors that indicate that speakers are sometimes planning ahead although they may not notice it (Redford, 2015). This planning phase is considered important both at the sentence and discourse levels. For example, we may decide to use the "In order to ..., he ..., because ..." structure before telling a long story. Therefore, it is meaningful to explore the benefits for a neural language generation model to have a such a planning phase.

In this work, we are interested in letting the model plan the sentence structure before producing words. If this can be done, then we may be able to select a structure beforehand, or obtain multiple sentences with diverse structures. Such features are useful for language generation tasks such as machine translation, as users may desire to know how a sentence can be translated with different structures. Therefore, we focus on machine translation task when developing our models.

In contrast to human, a neural machine translation (NMT) model does not have a planning phase when it is asked to generate a sentence. One can argue that the planning is performed in the hidden layers. However, as a source sentence can have multiple valid translations with different structures. When we use the cross-entropy loss to train the model, it has to assign probabilities to all possible translations with different structures. Therefore, the structural information remains probabilistic but not deterministic in the neural network. In other words, the model is unaware of the "big picture" of the output sentence before translation. Consequently, we can not control the sentence structure beforehand.

To obtain diverse sentences, conventional NMT models allow one to sample translations from the $N$-best list produced by beam search, but the sampled translation results usually share similar syntactic structures.

One naive approach to explicitly control the output structure is to prefix translations with their structural (e.g., part-of-speech) tags. In this case, the NMT model has to predict the tags before generating words according to these tags. Thus, we can obtain structural diverse translations by using different tags. This approach is referred to as *tag planning*. In our experiments, we found this approach results in significant degradation in translation quality because errors in the planning phase directly

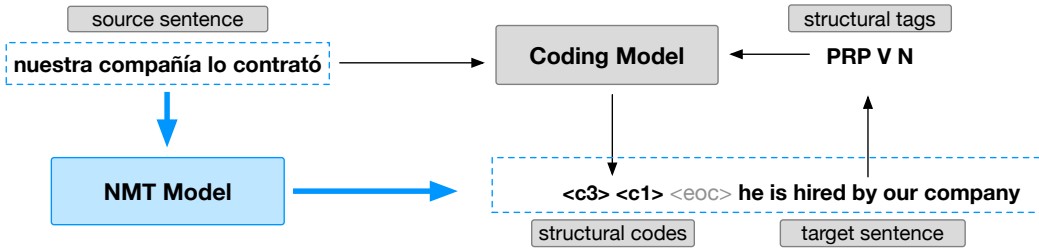

Figure 1: Proposed approach for planning a sentence structure with discrete codes. A target sentence is parsed to obtain its structural tags. Then, based on the tags and source sentence, the coding model produces multiple codes containing information about the structural choice. Finally, an NMT model is trained based on the source and target sentences with the structural codes as prefixes.

affect the correctness of the translation. To overcome this problem, we propose learning a discrete structural representation (i.e., discrete codes) that capture structural variations without negatively affecting the translation quality.

Fig. 1 shows our proposed two-phase approach for training the NMT model. In the first phase, we parse target sentences to obtain their structural tags. In this work, we use simplified POS tags to represent the coarse structure of a sentence. Next, we design a coding model that encodes information about the structural variations into discrete codes. In the second phase, we prefix each target sentence in the training data with the structural codes produced by the coding model. As shown in Fig. 1, we use a special token "⟨eoc⟩" to denote the end of the codes. Finally, we train an NMT model on the enhanced dataset; the model architecture does not require any modification.

To maintain translation performance, our coding model eliminates partial information from the structural tags that is unambiguous when the source sentence is given. For example, for the Spanish sentence shown in Fig. 1, it is unambiguous that the translation will contain a noun, a pronoun, and a verb. Such obvious information does not require planning, and the translation quality may be impacted seriously if it is predicted wrongly. In this example, the structural choice between "he is hired by our company" or "our company hired him" is ambiguous. In this work, the coding model is designed to capture only the ambiguous part of the sentence structure when the source sentence is known. We call this approach *discrete planning*, as the planning component is actually the artificially created discrete codes.

Please note that the planning component can also be a continuous latent vector, which requires a discriminator to train the model in order that the latent captures disentangled structural information. However, such a discriminator is difficult to implement. A detailed discussion is given in related work. Our approach of learning discrete structural representation has the advantage of simplicity, and no modification to the NMT model is required.

The contributions of this work can be summarized as follows:

1. We propose a novel approach to plan the sentence structure with a discrete structural representation. By carefully learning the discrete codes to capture pure structural variation, the translation quality remains intact.

2. We are able to obtain drastically different translations by manipulating the codes. To qualitatively evaluate the diversity, we propose a structural diversity metric. The evaluation results show that our approach is capable of producing sentences with highly diverse structures.

## 2    LEARNING STRUCTURAL CODES

This section discusses the motivation and methodology for learning discrete structural codes. First, we describe a naive approach that extracts the global sentence structure from part-of-speech (POS) tags. Then, we describe a coding model that learns discrete structural codes and the advantages of doing so.

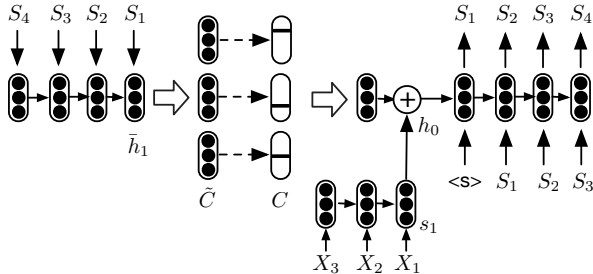

Figure 2: Architecture of code learning model. Discretization bottleneck is indicated by dashed lines.

## 2.1 EXTRACTING GLOBAL SENTENCE STRUCTURES

To plan the structures of translations, we need to obtain the structural representation of a sentence. However, it is extremely difficult for a neural network to automatically realize and extract the sentence structure. Therefore, we consider utilizing the results of POS tagging or syntactic parsing. However, most parsing results including POS tags contain detailed structural information, which is not desirable. For example, "destroy" and "break down" are interchangeable, but they have different POS tags. Such minor differences do not affect the global structure of a sentence. Therefore, we need a structural representation that captures the sentence structure at a coarse level.

In this work, we extract such a global sentence structure through a simple two-step process that simplifies the POS tags:

1. Remove all tags other than "**N**", "**V**", "**PRP**", "**,**" and ".". Note that all tags begin with "N" (e.g. **NNS**) are mapped to "**N**", and tags begin with "V" (e.g. **VBD**) are mapped to "**V**".

2. Remove duplicated consecutive tags.

The following list gives an example of the process:

```
    Input: He found a fox behind the wall.
POS Tags: PRP VBD DT NN IN DT NN .
  Step 1: PRP V N N .
  Step 2: PRP V N .
```

It should be noted that other parse results can also be potentially useful to represent the global structure; this is left for future works to explore. In our experiments, we found simplified POS tags to be effective and convenient for implementation.

## 2.2 CODE LEARNING

Although we can directly condition the word prediction using the structural tags, we found that errors during structural prediction significantly degraded the translation performance. To mitigate their impact, we propose learning a set of discrete codes that capture only the ambiguous structural choice when the source sentence is given.

**Code Encoder** Given the sentence pair $(X, Y)$, let the structural tags of the target sentence be $S_1, ..., S_T$, the encoder computes the logits of $N$ softmax vectors:

$$\bar{h}_t = \text{LSTM}(\text{E}(S_t), \bar{h}_{t+1}; \phi_e) , \tag{1}$$

$$[\tilde{C}_1, ..., \tilde{C}_N] = \text{softplus}(f(\bar{h}_1; \phi_h)) , \tag{2}$$

where the tag sequence $S_1, ..., S_T$ is firstly encoded using a *backward* LSTM (Hochreiter & Schmidhuber, 1997). $\text{E}(\cdot)$ denotes an embedding function, and $f(\cdot)$ denotes a linear transformation. Then, we compute a set of continuous vectors $\tilde{C}_1, ..., \tilde{C}_N$, which are latterly discretized into approximated one-hot vectors $C_1, ..., C_N$ using Gumbel-Softmax reprarameterization trick (Jang et al., 2016; Mad-

dison et al., 2016) as:

$$C_i = g(S, \epsilon; \phi)_i \tag{3}$$

$$= \text{softmax}_\tau(\log \tilde{C}_i + \text{gumbel}(\epsilon)) . \tag{4}$$

Here, $\tau$ is the temperature of the softmax. The computation of Gumbel-softmax relies on the Gumbel noise, which is defined as $\text{gumbel}(\epsilon) = -\log(-\log(\epsilon))$. As both $\tilde{C}_i$ and $\epsilon$ are vectors in Eq. 4, the $\log(\cdot)$ here is an element-wise operation. The random variable $\epsilon$ is sampled from a uniform distribution $U(0, 1)$. In Eq. 3, $g(S, \epsilon; \phi)$ represents the encoder function, which is used to produce the codes once its parameter $\phi$ is trained.

**Decoder** In the decoder, we combine the information from $X$ and $C$ to initialize a decoder LSTM that sequentially predicts the structural tags $S_1, ..., S_T$:

$$s_t = \text{LSTM}(\text{E}(X_t), s_{t+1}; \theta_x) , \tag{5}$$

$$h_0 = f([C_1, ..., C_N]; \theta_h) + s_1 , \tag{6}$$

$$h_t = \text{LSTM}(\text{E}(S_{t-1}), h_{t-1}; \theta_d) , \tag{7}$$

where $[C_1, ..., C_N]$ is a concatenation of $N$ Gumbel-softmax code vectors. Note that only $h_t$ is computed with a *forward* LSTM. Finally, we predict the probability of emitting each tag $S_t$ with

$$P(S_t | S_{<t}, X, C) = \text{softmax}(f_{\text{out}}(h_t; \theta_{\text{out}})) . \tag{8}$$

**Loss Function** The architecture of the code learning model is depicted in Fig. 2, which can be seen as a sequence auto-encoder with an extra context input $X$ to the decoder. The loss function for optimizing the entire model has the following form given one training pair:

$$L(\phi, \theta) = \mathbb{E}_{\epsilon \sim U(0,1)} \Big[ -\log P(S | X, g(S, \epsilon; \phi); \theta) \Big] , \tag{9}$$

$$= \mathbb{E}_{\epsilon \sim U(0,1)} \Big[ \sum_{t=1}^{T} -\log P(S_t | S_{<t}, X, g(S, \epsilon; \phi); \theta) \Big] . \tag{10}$$

The codes are limited to have only a few bits of information capacity in our experiments. In order to minimize the loss function, the codes produced by the encoder $g(\cdot)$ has to capture the structural information contained in $S$ that cannot be inferred by $X$.

### 2.3 NMT with Structural Planning

Once the coding model is trained, the Gumbel-softmax code vectors $C_1, ..., C_N$ shall be very close to one-hot vectors. We can extract $N$ discrete codes by applying *argmax* on the softmax vectors, and prefix the target sentences in the training data with the codes, resulting in $(X, C; Y)$ pairs. We connect the codes and target sentences with a "⟨eoc⟩" token.

To evaluate the performance of structural planning with discrete codes, we train a regular NMT model with the modified dataset. The codes are removed from the translation results after decoding. Similarly, we can evaluate the tag planning model using $(X, S; Y)$ pairs.

## 3 Related Work

Our work learns a discrete representation of structure and leverages it obtain diverse translations, which is related to topics including style transfer and diverse language generation. Though differing in purpose, our approach is also related to NMT models with syntactic constraint.

**Style Transfer for Natural Language** As our task learns the structural representation, the problem setting is closely related to style transfer for language generation tasks. Hu et al. (2017) simultaneously learns a disentangled representation along with entangled representations within the same latent vector. The disentangled representation is trained with a style-specific discriminator. Shen et al. (2017) learns a cross-aligned auto encoder to tackle the style transfer problem without parallel text data. Prabhumoye et al. (2018) utilizes back-translation to create a style agnostic representation of sentence. Then multiple style transfer decoders is trained to generate sentences in specific styles

according to the latent representation. The parameters are optimized with style classifiers using adversarial loss.

The main difference between our task and previous works in style transfer is in the objectives. In the aforementioned works, the styles have finite categories. For example, the label for sentiment style can only be "positive" or "negative". In our task setting, the objective is to learn a representation that captures the global sentence structure, which has infinite variations.

Theoretically, we can still train a variational auto-encoder (VAE) that learns a categorical latent code or a continuous latent variable with a Gaussian prior to capture the structure. We can extend the VAE model described in Hu et al. (2017) with an enhanced discriminator $q_D(c|x)$ to map the global structure of a sentence $x$ to the space of a latent variable $c$, and ensure that sentences with similar structures are mapped to same or close latent representations. However, as the structural information is highly entangled with the utterances, such a discriminator is fairly difficult to implement.

**Diverse Language Generation**  As generating diverse text has its application in tasks such as conversation generation, several existing works are presented to improve the diversity of language generation. Similar to Hu et al. (2017), Dai et al. (2017) generates diverse image captions with a conditional GAN, whereas Jain et al. (2017) generates diverse questions with a VAE.

Some creative approaches are also proposed on this topic. Li et al. (2016) improves the diversity of generation by modifying the scoring function in beam search. A hyperparameter that controls the diversity is optimized by reinforcement learning. Xu et al. (2018) learns $K$ shared decoders, each is conditioned on a trainable pattern embedding. In each iteration, only the decoder with lowest cross-entropy receives training signal.

Existing works on language generation care about using creative vocabulary. Both Li et al. (2016) and Xu et al. (2018) measure the diversity with corpus-level metrics. The former work evaluates the number of unique $n$-grams in the generated text, whereas the latter one measures the divergence of word distributions produced by different style-specific decoders. Our approach differs from these works by putting the focus on improving the structural diversity instead of choice of word.

**NMT with Syntactic Constraint**  To our knowledge, there are no existing works control the sentence structure ahead of translation. However, our approach is related to NMT models under syntactic constraint. Stahlberg et al. (2016) syntactically guides the NMT decoding using the lattice produced by Hiero, a statistical machine translation system allowing hierarchical phrase structure. Eriguchi et al. (2017) parse a dependency tree and combine the parsing loss with the original loss, which improves the syntactic trace of translations in a soft way.

Similar to our approach, the following works also enhance the target sentence with extra structural tags. Nadejde et al. (2017) interleaves CCG supertags with normal output words in the target side. Instead of predicting words, Aharoni & Goldberg (2017) trains an NMT model to generate linearized constituent parse trees. Wu et al. (2017) proposed a model to generate words and parse actions simultaneously. The word prediction and action prediction are conditioned on each other. These works focus on improving the syntactic correctness, whereas our work focuses on structural diversity.

## 4 EXPERIMENTS

### 4.1 EXPERIMENTAL SETTINGS

We evaluate our models on two machine translation datasets: IWSLT14 German-to-English dataset (Cettolo et al., 2014) and ASPEC Japanese-to-English dataset (Nakazawa et al., 2016). These datasets contain 178K and 3M bilingual pairs, respectively. For the IWSLT14 De-En dataset, we apply the Moses toolkit (Koehn et al., 2007) to tokenize both sides of the corpus. For the ASPEC Ja-En dataset, we use the Moses toolkit to tokenize the English side and Kytea (Neubig et al., 2011) to tokenize the Japanese side.

After tokenization, we apply byte-pair encoding (Sennrich et al., 2016) to segment the texts into subwords, forcing the vocabulary size of each language to be 20K and 40K for the IWSLT dataset and ASPEC dataset, respectively.

For evaluation, we report *tokenized BLEU* with Moses script. We concatenate all five TED/TEDx development and test corpora to form a test set containing 6750 pairs in the IWSLT14 dataset. For the ASPEC dataset, as we truecase the English texts in the preprocessing phase, we detruecase the results before evaluating the scores. For planning models, we remove the structural codes (or tags) from the results before executing the evaluation script.

## 4.2 EVALUATION OF CODING MODEL

**Training Details:** In the coding model, all hidden layers have 256 hidden units. The model is trained using Nesterov's accelerated gradient (NAG) (Nesterov, 1983) for a maximum of 50 epochs with a learning rate of $0.25$. The parameters with the lowest validation loss are selected for evaluation. During training, the temperature $\tau$ in the Gumbel-Softmax is fixed to $1.0$. We also tried to anneal the Softmax temperature; however, this did not result in any performance improvement.

When training the coding model, we test different settings of code length $N$ and the number of code types $K$. Following the code setting, the coding model will learn $N$ Gumbel-Softmax vectors, each having a size of $K$. In this case, the information capacity of the codes will be $N \log K$ bits.

**Evaluation Results:** We evaluate the code quality from two aspects. First, we measure the reconstruction accuracy that indicates how accurately we can recover the original tag sequence from the codes. Second, to determine the difficulty of predicting the correct codes given the input sentence, we add an extra Softmax layer after the last hidden vector of the $X$ encoder to predict the generated codes. Although the additional Softmax layer is trained simultaneously with the coding model, we disconnect the backpropagation path to ensure that the code prediction loss does not affect other components.

| code setting | capacity | IWSLT14 De-En | | ASPEC Ja-En | |
|:---:|:---:|:---:|:---:|:---:|:---:|
| | | tag accuracy | code accuracy | tag accuracy | code accuracy |
| N=1, K=4 | 2 bits | 27% | 63% | 35% | 40% |
| N=2, K=2 | 2 bits | 23% | 67% | 27% | 55% |
| N=2, K=4 | 4 bits | 35% | 41% | 58% | 25% |
| N=4, K=2 | 4 bits | 22% | 44% | 18% | 79% |
| N=4, K=4 | 8 bits | 44% | 27% | 14% | 58% |

Table 1: A comparison of different code settings on IWSLT14 De-En and ASPEC Ja-En datset. For each code setting, we report the accuracy of recovering the structural tag sequence using the codes (tag accuracy), and the accuracy of predicting the codes using the source sentence (code accuracy).

Table 1 shows the evaluation results for different code settings. We can see a clear trade-off between the reconstruction accuracy and code prediction accuracy, especially on IWSLT14 dataset. When the code has more capacity, it can recover structural tags more accurately; however, this results in lower predictability.

Generally, we want the codes to contain more accurate information about the sentence structure. However, the translation quality may be negatively impacted if it is too difficult for the NMT model to guess a good code. Therefore, we choose the code settings $N = 1, K = 4$ and $N = 2, K = 4$ for the following experiments as these provide a balanced trade-off.

## 4.3 EVALUATION OF NMT MODELS

**Model Architecture:** To create a strong NMT baseline, we use two layers of bidirectional LSTM encoders with two layers of LSTM decoders in the NMT model. The hidden layers have 256 units for the IWSLT De-En task and 1000 units for the ASPEC Ja-En task. All parameters are initialized using the Xavier method (cite). We apply Key-Value Attention (Miller et al., 2016) to the first decoder layer, and we use residual connection (He et al., 2016) to combine the hidden states in two decoder layers. Dropout is applied everywhere outside of the recurrent function with a drop rate of $0.2$ . Before the final Softmax layer, we insert another full-connected layer that has 600 hidden units.

| Dataset | Model | BLEU(%) | | | |
|---|---|---|---|---|---|
| | | BS=1 | BS=3 | BS=5 | BS=10 |
| IWSLT14 De-En | baseline NMT | 27.90 | 29.26 | 29.52 | 29.34 |
| | tag planning | 19.62 | 20.44 | 20.55 | 20.66 |
| | discrete plan (N=1, K=4) | 28.02 | 29.23 | 29.51 | 29.60 |
| | discrete plan (N=2, K=4) | **28.35** | **29.59** | **29.78** | **29.83** |
| ASPEC Ja-En | baseline NMT | **23.92** | 25.08 | 25.26 | 25.48 |
| | tag planning | 17.55 | 18.54 | 18.73 | 19.02 |
| | discrete plan (N=1, K=4) | 23.42 | 25.29 | 25.32 | 25.59 |
| | discrete plan (N=2, K=4) | 22.79 | **25.53** | **25.69** | **25.88** |

Table 2: Performance evaluation for NMT models trained with different approaches. The BLEU scores are reported for various beam sizes (BS).

**Training Details:** To train the NMT models, we also use the NAG optimizer with an initial learning rate of 0.25, and it is annealed by a factor of 10 if no improvement of loss value is observed in 20K iterations. One batch contains 128 sentence pairs, and the computation is distributed to eight GPUs. The best parameters are chosen on a validation set.

Note that all evaluated models share the same architecture, and they only differ in terms of the target side of training data. For the tag planning model and discrete planning model, the target sentences are prefixed by tags and codes, respectively.

**Evaluation Results:** Table 2 shows the resultant BLEU scores of different models, which indicates that our proposed planning approach does not degrade the translation performance in both translation tasks. In contrast, when we use the structural tags to directly control the sentence structure, a significant drop in BLEU scores is observed. Such a negative impact on translation quality is undesirable for structural planning.

In both De-En and Ja-En tasks, we can also observe a minor increase in performance over the strong baseline when using beam search (BS $> 1$). The improvement may result from being able to simultaneously explore drastically different candidates. A recent study (Li et al., 2016) also showed that beam search performance can be improved by increasing the diversity of candidates.

When we compare the resultant translation performance of different code settings, we found the translation scores are not worsen when using a longer code sequence ($N = 2$), comparing to planning with only one code ($N = 1$). This is benefited by avoiding encoding predictable structural information when learning the codes, thereby minimizing the influence of code prediction errors.

## 4.4 EVALUATION OF STRUCTURAL DIVERSITY

| Dataset | Model | DP (%) | POS-DP (%) |
|---|---|---|---|
| IWSLT14 De-En | baseline NMT | 35.99 | 27.72 |
| | discrete plan (N=1, K=4) | **39.79** | **36.51** |
| ASPEC Ja-En | baseline NMT | 25.89 | 20.28 |
| | discrete plan (N=1, K=4) | **52.10** | **45.42** |

Table 3: Evaluation of discrepancy scores (DP) for the baseline model and the discrete planning approach. We also report the discrepancy scores of the part-of-speech tags of candidates (POS-DP).

In our approach, the codes are learned to determine the target structure together with the input. A single code is not guaranteed to be aligned to a specific style. Therefore, we can not evaluate the diversity with the divergence between word distributions as Xu et al. (2018) does. In order to qualitatively evaluate the diversity of generated translations, we propose to use a simple BLEU

| Input | | damit sind wir zu viele . (German) |
|---|---|---|
| **BS** | **1** | `so we're too many .` |
| | **2** | `so we are too many .` |
| | **3** | `that's where we are too many .` |
| **PL** | **1** | **`<c4> <c1> <eoc>`** `we're too many of them .` |
| | **2** | **`<c1> <c1> <eoc>`** `that's where we are too many .` |
| | **3** | **`<c3> <c1> <eoc>`** `so , we're too many .` |
| **Input** | | AP NO KATEI NI TSUITE NOBETA . (Japanese) |
| **BS** | **1** | `process of the AP is described .` |
| | **2** | `a process of AP is described .` |
| | **3** | `reaction of AP is described .` |
| **PL** | **1** | **`<c3> <c2> <eoc>`** `the process of AP is described .` |
| | **2** | **`<c1> <c2> <eoc>`** `this paper describes the process of AP .` |
| | **3** | **`<c4> <c2> <eoc>`** `here was described on process of AP .` |

Table 4: A comparison of translation candidates produced by beam search (BS) and discrete planning (PL) in IWSLT14 De-En and ASPEC Ja-En task

discrepancy metric. Suppose $Y$ is a list of candidate translations, we compute the discrepancy with

$$\text{DP}(Y) = \frac{1}{|Y|(|Y|-1)} \sum_{y \in Y} \sum_{y' \in Y, y' \neq y} 1 - \Delta(y, y'), \quad (11)$$

where $\Delta(y, y')$ computes the BLEU score of two candidates. The equation computes the mean value of 1 - BLEU between all candidate pairs. We get a high discrepancy score when all candidates are very different from each other. As Eq. 11 evaluates for one data point, we further average the discrepancy scores over the test dataset.

In order to see the structural diversity, we further propose a POS-DP metric, in which $\Delta(y, y')$ computes the BLEU scores using the POS tags of two candidates. Therefore, lexical diversity is completely omitted in the scores. The POS-DP metric is expected to reflect the structural diversity.

We report the discrepancy scores in Table 3. We first apply beam search on the baseline model with a beam size of 4 to obtain four candidate translations for each input, and then measure the baseline discrepancy scores. Then we sample candidate sentences from a discrete planning model trained with a code setting $N = 1, K = 4$. As there are only four possible codes, we sample four translations, each conditioned on a unique code. Then we measure the mean discrepancy on the sampled translations.

We can see that the proposed method achieves higher discrepancy scores, which means that the candidate translations are less similar to each other. A larger gap in diversity is observed when evaluating with the POS-DP metric. It indicates that the proposed method creates more diversity in sentence structures.

## 5 Qualitative Analysis

### 5.1 Sampling Translations with High Diversity

Instead of letting the beam search to decide the best structural codes, we can also choose the codes manually. Specifically, we use beam search to obtain three code sequences with highest scores. Then for each code, we sample the translation result conditioned on it. Table 4 gives some samples of the candidate translations produced by the model when conditioning on different structural codes, compared to the candidates produced by beam search.

We can see that in both tasks, the candidate translations produced by beam search algorithm has only minor difference in grammatical structure. In contrast, the translation results sampled with discrete

| | |
|---|---|
| **Input** | KAIHATSU JOUKYOU WO SHOUKAI SHI TA . (Japanese) |
| **Condition 1** | `the features of this program are explained .` |
| **Result 1** | `the development situation is introduced .` |
| **Condition 2** | `they explained features of this program .` |
| **Result 2** | `they introduced the development situation .` |
| **Condition 3** | `this paper explains the features of this program .` |
| **Result 3** | `this paper introduces the development situation .` |

Table 5: Constraining structures of translations with condition sentences

planning have drastically different structures. We can observe the diversity more significantly in Japanese-English task because the word order of translations is less relevant to source sentences in this task. We can also notice that as a result of learning from structural tags, the discrete codes control the sentence structure rather than the choice of word.

The results in Table 4 show that the proposed method can be useful for sampling paraphrased translations with high diversity. Such property can also benefit other language generation tasks such as conversation generation.

## 5.2 TRANSLATING UNDER STRUCTURAL CONSTRAINT

Sometimes, we want the NMT model to generate a translation with a specific sentence structure. However, after we encode the structural information into discrete codes, it is difficult to clearly tell what sentence structure a specific code sequence indicates. This problem can be overcome by encoding a known sentence into codes in the hope that the NMT model can generate a translation with a similar structure when conditioned on the codes.

Table 5 shows some results obtained when using a known sentence to constrain the structure of translation in the Japanese-English translation task. The translation results are sampled using the discrete codes extracted from the conditional sentences. Each translation shows a global sentence structure similar to the condition sentence, although they have different utterances and local structures. The results confirm that our coding model captures the variations among different structures.

## 6 CONCLUSION

In this paper, we add a planning phase to neural machine translation, which plans the global sentence structure before decoding words. The structural planning is achieved by using some discrete codes to control the structure of output sentences. The discrete codes are learned to capture the structural variation of a translation. We prefix all target sentences in the corpus and use the enhanced training data to train an NMT model. During translation, the model first generates the codes, then output the words conditioned on the structural constraint imposed by the codes.

To learn the codes, we design an end-to-end neural network with a discretization bottleneck to encode the information of structural tags into some one-hot vectors. Experiments show that the proposed method can avoid degrading the translation performance. We also confirm the effect of the structural codes, by being able to sample translations with drastically different structures using different codes. We can also constrain the structure of a translation by using the codes extracted from a known sentence.

Our approach can be used in other language generation tasks, such as conversation modeling. Also, the discrete coding model can be extended to learn discrete representations of other factors of sentences. For example, one can learn the discrete codes to capture the logical order of a paragraph. In this case, the results of discourse parsing can be used instead of structural tags. By manipulating the codes in the planning phase, we may control the logical flow of the output sentences.

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

# A   ANALYSIS OF CODE EFFICIENCY

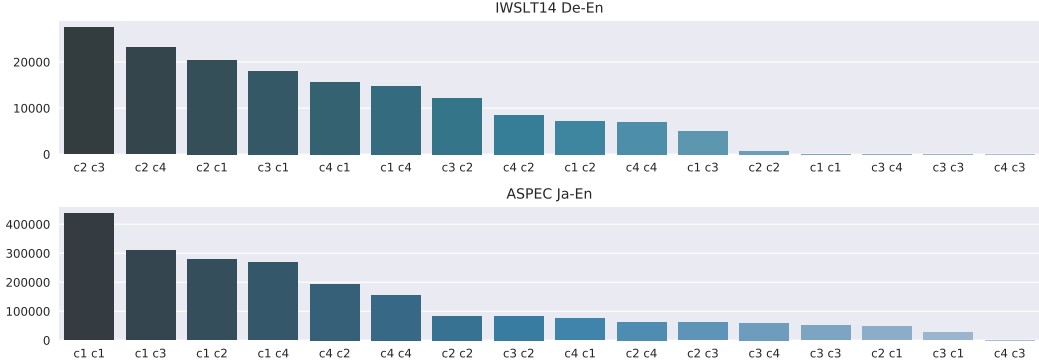

Figure 3: Visualization of code distribution in IWSLT14 De-En task and ASPEC Ja-En task

In Fig. 3, we report the distribution of the structural codes learned for all target sentences in both tasks. In each plot, the Y-axis indicates the total number of sentences that correspond to a specific structural code. In general, the coding model is able to distribute available discrete codes to different sentences in the training data. However, we can still observe that four codes in IWSLT14 and one code in ASPEC task are not assigned to any sentence. Thus, there is room for further improving the coding model to exploit the full capacity of the codes, such as using an improved discretization method (Kaiser et al., 2018).

