# OpenReview forum: "Discrete Structural Planning for Generating Diverse Translations"
_ICLR.cc/2019/Conference_

### Official Review · AnonReviewer3 · 2018-10-31
**Interesting approach to translation diversity, but experiments somewhat lacking and details missing**

**Rating:** 5
**Confidence:** 3

**Review:**

The authors propose modeling structural diversity of translations by conditioning the generation on both the source sentence and a latent encoding of the overall structure (captured by simplified part-of-speech tags). Specifically, they first train a conditional autoencoder to learn a latent code optimized towards reconstructing the tag sequence. They then prefix the inferred latent code to the target sentence before generation. A diversity metric which measures pairwise BLEU scores between beam items is also proposed. Experiments show that the latent codes lead to greater structural diversity as well as marginally improved translation results when combined with beam search.

Contributions
-----------------
A simple method for improving structural diversity.

The use of conditional autoencoding to capture structural ambiguity, while not in itself novel, could be interesting for other problems as well.

Experiments suggest that the method is rather effective (albeit only improving translation quality marginally)

I like the proposed discrepancy score based on pairwise BLEU scores.

Issues
---------
It is not clear if teacher forcing was used in the "tag planning" setting. If gold tag sequences were used during training there is a major train/test mismatch which would explain the dramatic drop in BLEU scores. If so, this is a major issue, since the authors claim that as the motivation for the use of discrete latent codes. To make the "tag planning" setting comparable to the latent code setting, you would need to train the tag prediction model first and then condition on predicted tags when training the translation model (potentially you would need to do jack-knifing to prevent overfitting as well).

It is unfortunate that there is no empirical comparison with the most closely related prior work, in particular Li et al. (2016) and Xu et al. (2018), which are both appropriately cited. As it stands it is not possible to tell which of these approaches is most useful in practice.

No details are provided on the tagset used and what system is used to predict it, or to what degree of accuracy.

Having a fixed number of codes regardless of sentence length seems like a major shortcoming. I would urge the authors to consider a variable coding length scheme, e.g., by generating codes autoregressively instead of with a fixed number of softmaxes. It would also be interesting to break down the numbers in table 1 with respect to sentence length.

Minor issues
-----------------
Citation for the Xavier method is missing.

Notation is somewhat hard to follow. Please add a few sentences describing it and make sure it is consistent.

There are many grammatical errors. Please make sure to proofread!

"Please note that the planning component can also be a continuous latent vector, which requires a discriminator to train the model in order that the latent cap." What does this mean?

---

> ### Author Response · Authors · 2018-11-12
> **Response to AnonReviewer3**
>
> Q1 It is not clear if teacher forcing was used in the "tag planning" setting
>
> Yes, as the decoder part of the coding model is a language model, it is teacher forcing for the “tag planning” approach. Please note that it is the same situation for the “discrete planning“ approach.
>
> The core problem is that the tags contain  critical information about the translation not only in structural level. For example, if the NMT model fails to generate a NN tag for sentence “A dog is playing”, then the final translation result will be very wrong. Therefore, the tag predicting part has to be fairly accurate in order to eliminate the influence to the translation performance.
>
> By learning the codes to capture things that are not obvious given the source sentence, this problem is solved. The codes will not contain information about whether there is a NN tag (if it is an obvious thing) but the order of the tags.
>
> Q2 no empirical comparison with the most closely related prior work
>
> Indeed, the decoding approach of Li et al. (2016)  is easy to implement, we are going to include the results in the paper. However, as those methods are trying to increase the diversity on the choice of words,  it is unfair to evaluate them with the structural diversity metric.
>
> As the examples in the qualitative analysis show, our approach only produce diversity in structural level. Therefore, our approach is suitable for translation systems to generate multiple translation candidates, where the users are not expecting the system to use “surprisingly creative” words.
>
> Q3 No details are provided on the tagset used and what system is used to predict it
>
> For the POS tagging part, we are using nltk.pos_tag .
>
> Q4 Having a fixed number of codes regardless of sentence length seems like a major shortcoming.
>
> We actually consider this as the strength of the discrete coding approach, because :
>
> 1) using a fixed number of codes can potentially reduce the chance of error when predicting the codes.
>
> 2)  we can enumerate all candidate codes when they have a fixed number of sub codes.

---

> > ### Comment · AnonReviewer3 · 2018-11-12
> > **Fundamental issues remain**
> >
> > Thank you for your response.
> >
> > There seems to be a fundamental misunderstanding re: teacher forcing. It is not teacher forcing that is the issue, the issue is that when training with gold part-of-speech tag sequences the model can look into the future as the tag sequence is derived directly form the target side to be predicted. This issue is not as severe for the autoencoded discrete codes since these are predicted from the source side only. If the part-of-speech tag sequence was predicted in the same way, there would be no time travel effect.
> >
> > Re: using a fixed number codes. Sure this makes the model slightly simpler, but it is a fundamental limitation since it cannot account for the structure of longer sentences.

---

> > > ### Author Response · Authors · 2018-11-13
> > > **Re: Fundamental issues remain**
> > >
> > > I understand your concern. You mean that translation quality may be improved if we remove the exposure bias on tags. I'm working to get the results in this scenario. I will first train a NMT model to predict the tag sequence from the source sentence, then concatenate the predicted tags and the target sentences to train a full NMT model.
> > >
> > > My concern is that if we do not use the reference tags when training the NMT, then the tags can not be used to condition the structure of translations.

---

> > > ### Author Response · Authors · 2018-11-21
> > > **Additional experiment results without teaching forcing on the tags**
> > >
> > > Hi, we conducted additional experiments with a model does not apply teacher forcing on the tags. This means we first train the model to predict tags. Then conditioning on the predicted tags, we  train the model to generate translations.
> > >
> > > In the additional experiments, we use standard transformer (6 encoders, 6 decoders) architecture on the large ASPEC Ja-En dataset (3M sentence pairs). Here are the quantitive results:
> > >
> > > Model 		BS=3		BS=5		BS=10
> > > Baseline		25.92		26.02		26.02
> > > code_plan	26.97		27.4		27.39
> > > tag_plan		27.32		27.6		27.75
> > >
> > > The first observation is that both planning approach improves the BLEU scores even more significantly with transformer. They are all very high scores.
> > >
> > > For the tag planning approach , we found that the accuracy of predicting the tag sequences is only 10.6%. Around 1/3 of the sentences are predicted to have the structure “N V N V N V N .”, which is wrong. As a result, the translation sentence does not follow the tag sequence it predicted.
> > >
> > > Example translation:
> > > N V N V N V N . <eoc> except for 3PCIS and three-phase reference signal generator , it is composed of the same parts as the conventional optical stripe range finder .
> > >
> > > We can see that the resultant translation is not related to the predicted structural tags. It implies that we are unable to control the structure of translation with the tags. In contrast, the planning approach with discrete codes works well with teacher forcing on the codes.

---

> > > > ### Comment · AnonReviewer3 · 2018-11-21
> > > > **Needs more work**
> > > >
> > > > In summary tag planning gives higher BLEU scores, but discrete latents allow controlling diversity?
> > > >
> > > > While these results are interesting, it seems to me that there are too many moving parts that needs to be rigorously investigated for the paper to be publishable. I therefore keep my score.

---

### Official Review · AnonReviewer2 · 2018-11-03
**Attacking an interesting but mostly solved problem with weak baselines and questionable ML techniques**

**Rating:** 4
**Confidence:** 5

**Review:**

The authors consider the problem of generating diverse translations from a neural machine translation model. This is a very interesting problem and indeed, even the best models lack meaningful diversity when generating with beam-search. The method proposed by the authors relies on prefixing the generation with discrete latent codes. While a good general approach, it is not new (exactly the same general approach that was used in the "Discrete Autoencoders for Sequence Models" [1] paper, https://arxiv.org/abs/1801.09797, for generating diverse translations, which is not cited directly but a follow-up work is cited, though without mentioning that a previous work has tackled the same problem). Also, the authors rely on additional supervised data (namely POS tags) which has no clear motivation and seems to cause a number of problems -- why not use a purely unsupervised approach when it has already been demonstrated on the same problem? Additionally, the authors compare to a weak translation baseline on small data-sets, making it impossible to judge whether the results would hold on a larger data-set. So the following ablations and comparison to baselines are missing:
* comparing with a stronger NMT architecture and larger data-set
* does the chosen discretization method matter? Other methods have been shown to strongly out-perform Gumbel-Softmax in this context, so a comparison would be in order.
* comparison to fully unsupervised latents and some other system, e.g., the system from [1] above

In the absence of these comparisons and with little novelty, the paper is a clear reject.

[Revision]

Greatly appreciate the answers provided by the authors. The Ja-En dataset is indeed much larger than I thought, so I increased my score. When the other points are addressed (as the authors say they will do) it may be a good paper -- but the review must stick to the submitted version, not a future one.

---

> ### Author Response · Authors · 2018-11-08
> **Response to AnonReviewer2**
>
> Thank you for spending time for reviewing my paper.
>
> Q1 comparing with a stronger NMT architecture and larger data-set
>
> As the ASPEC Ja-En dataset contains 3M bilingual sentence pairs, it shall not be considered as a small dataset.
>
> Also our baseline model  achieves a strong BLEU score with various techniques.  On IWSLT14,  our baseline model achieves a  BLEU score of 29.34 (beam size = 10). A recent facebook paper (Gehring et al.,  2017: A Convolutional Encoder Model for Neural Machine Translation) also reported scores on the exactly same training and test data. Their convolutional encoder model achieves a BLEU score of 29.9, and a deep convolutional setting achieves 30.4 BLEU. So we do not think  our baseline is a weak model.
>
> Q2 Other methods have been shown to strongly out-perform Gumbel-Softmax in this context, so a comparison would be in order
>
> We will also report the results with improved semantic hashing technique (Kaiser and Bengio, 2018: Discrete Autoencoders for Sequence Models)
>
> Q3 comparison to fully unsupervised latents and some other system
>
> The main point of this paper is to generate translations with drastically different structures. Previous work of diverse language generation are focusing on letting the model generate sentences with creative vocabularies but not structures.
>
> (Kaiser and Bengio, 2018)  shows some examples of diverse translations resulted by using the unsupervised latents, which are trained with improved semantic hashing. However, they did not evaluate the quality of the sampled diverse translations.
>
> We can indeed generate very diverse translation results if we are allowed to significantly degrade the translation quality.  However, such a performance degradation is not desirable in real products.
>
> Our approach preserve the translation quality by:
>
> 1) Using syntactic tags so that the utterances of the results will not be constrained.
> 2) Letting the codes contain only the target-side structural information that can not be predicted given the source sentence.

---

### Official Review · AnonReviewer1 · 2018-11-05
**Poorly motivated and confusing**

**Rating:** 2
**Confidence:** 5

**Review:**


This paper is not ready for publication in ICLR or most other venues. The model is poorly motivated, many modeling choices are confusing, and the experiments are not convincing.  I found much of the paper confusing. A (far from complete) sample:


§1 ¶1  What is this structure an example of? What sentence structures do you mean, concretely? Syntax? The introduction is very vague—I’m not convinced this is meaningful.

§1 ¶2-3 These paragraphs also vague.

§1 ¶5 Why is this approach naive? Is this a well-known method? There are no citations.

Fig.1 Very confusing: it looks like the target sentence, “structural tags” and “coding model” form a loop! This example is also confusing because the “structural tags” are non-sensical… they have no relation to this example sentence! I can’t tell if this is because they were made up without relation to the input sentence, or worse, that they’re an actual example from the data, in which case there is something very wrong with the tagger used in the “naive” experiments.

Sec. 2.1 What is the motivation behind the heuristics for the “two-step process that simplifies the POS tags”?

Sec 2.2. The description of the model is confusing. If I understand correctly, wehave training data for these “codes" (in the form of “simplified” POS tags), and a simple seq2seq model is the obvious first thing to try. Most of the choices that deviate from this (e.g. use of Gumbel-softmax, also confusingly called “softplus” in Eq. 2) are never explained.

Sec. 3 The related work is a laundry list of papers, explained without relation to the current paper. It simply gets in the way of the rest of the paper and isn’t needed.

Table 1. I’m not sure what the code accuracy tells us. It’s also unclear to me what is means to “reconstruct” the “original tag sequence” from the codes, esp. given the description in Sec 2.1.

Table 2. Given the minor differences in these numbers and the confusing description of the model and training process, I am skeptical of these numbers, which look quite a bit like noise. Note that the use of four columns corresponding to different beam sizes is misleading… this makes it look as if there are four separate experiments for each condition, but this is not really true, we expect these scores to correlate across different beam sizes, so seeing the bold numbers at the bottom of each column does not add substantial information.

Table 4. These are interesting, but it seems like a possibly natural consequence of adding a noisy sequence of characters to the beginning of the decoded sequence; I’m not convinced that the sequences mean anything per se, but it’s a bit like adding some random noise to the decoder state before generating the word sequence.

5.1 “Instead of letting the beam search decide the best … we use beam search to obtain three code sequences with highest scores.” I’m confused: what is the difference?

---

> ### Author Response · Authors · 2018-11-07
> **Response of AnonReviewer1**
>
> Thank you for spending time reviewing my paper.
>
> The main points of this paper are misinterpreted in the comments, here are the responses for them.
>
> Q1  What is this structure an example ?
>
> Grammar structure.
>
> Q2  Why is this approach naive?
>
> No modification to the model is required.
>
> Q3  Fig.1 This example is also confusing because the “structural tags” are non-sensical
>
> As the caption tells, it is the illustration of the proposed approach. “PRP V N” is the extracted structural tags for the example sentence. Please see section 2.1 for details.
>
> Q4  What is the motivation behind the heuristics for the “two-step process that simplifies the POS tags”?
>
> Extracting global sentence structures (title of section 2.1)
>
> Q5  The description of the model is confusing. A simple seq2seq model is the obvious first thing to try.
>
> A simple seq2seq model can not learn discrete codes. AnonReviewer1 failed to understand why a discretization method is required here. The Gumbel-softmax bottleneck allows the coding model to encode structural information into discrete codes.
>
> Q6  The related work is a laundry list of papers, explained without relation to the current paper.
>
> This comment is a false statement.  In the end of every parts of the related work, we are discussing the relation with this work.
>
> Q7 It’s also unclear to me what is means to “reconstruct”
>
> The sequence auto-encoder encodes the “tag sequence” into discrete “codes” and “reconstruct” the “original tag sequence”.
>
> The code accuracy shows how accurate can the codes be predicted given the source sentence.
>
> Note that the auto encoder produces the codes using the target-side structural tags. So after training the coding model, we have no idea whether the NMT model can correctly predict the codes only based on the source-side information.
>
> To reveal the predictability of the codes, we train an independent neural model  simultaneously with the coding model to predict the codes based on the source sentence, and report the accuracy.
>
> Q8 Given the minor differences in these numbers ...
>
> > Evaluation Results: Table 2 shows the resultant BLEU scores of different models, which indicates that our proposed planning approach does not degrade the translation performance in both translation tasks.
>
> As described in the paper, the numbers show that our approach of generating diverse translations does not significantly hurts the translation quality. This evaluation is important  as there is a trade-off between diversity and translation quality.
>
> Q9  it seems like a possibly natural consequence of adding a noisy sequence
>
> If it is the consequence of adding noise, how to explain the results in Table 5?
>
> Q10  “Instead of letting the beam search decide the best … we use beam search to obtain three code sequences with highest scores.”
>
> Using beam search, we can either adopt  only the best result, or retrieve a N-best list. In section 5.1, we use beam search to obtain top-3 code sequences, and generate sentences following each code sequence.

---

### Meta-Review · Area_Chair1 · 2018-12-14

**Confidence:** 4
**Recommendation:** Reject

**Metareview:**

This paper introduces a planning phase for NMT.  It first generates a discrete set of tags at decoding time, and then the actual words are generated conditioned on those tags.  The idea in the paper is interesting.

However, the paper's experimental settings could improve by comparing on larger datasets and also using stronger baselines.  The writing could also improve -- why were only the few coarse POS tags used?  Have the authors tried a larger set?  I think without such controlled comparisons, it would be hard to understand why only those coarse tags are used.

The reviewers express concern about some of the above issues and there is consensus that the paper should be improved for acceptance at a venue like ICLR.